# Active Aging and Smart Public Parks

**DOI:** 10.3390/geriatrics8050094

**Published:** 2023-09-22

**Authors:** João Boavida, Hande Ayanoglu, Cristóvão Valente Pereira, Rodrigo Hernandez-Ramirez

**Affiliations:** 1Unidade de Investigação em Design e Comunicação, Faculdade de Design, Tecnologia e Comunicação, UNIDCOM/IADE, Universidade Europeia, IADE, Av. D. Carlos I, 4, 1200-649 Lisboa, Portugal; rodrigo.ramirez@universidadeeuropeia.pt; 2Centro de Investigação e Estudos em Belas-Artes, Faculdade de Belas-Artes, Universidade de Lisboa, Largo da Academia Nacional de Belas-Artes, 1249-058 Lisboa, Portugal; c.pereira@belasartes.ulisboa.pt

**Keywords:** active aging, age-friendly, smart cities, smart public parks

## Abstract

The global population is aging, with the percentage of people over 60 expected to rise from 12% to 22% and 33% residing in developed countries. However, most cities lack the appropriate infrastructure to support aging citizens in active aging and traversing the urban landscape, negatively impacting their quality of life. Studies have shown that public parks and green spaces can contribute to a higher quality of life and wellbeing. Also, smart cities are intended to improve the wellbeing and health of their inhabitants. However, most solutions are typically implemented indoors and tend to overlook the needs of older adults. A smart city should consider the increasing rate of aging and give more importance to outdoor environments as a key aspect of quality of life. The article’s main purpose is to provide a comprehensive background to understand the current knowledge on smart public parks and highlight the significance of new research in the field to promote active aging. The article is expected to inspire new research ideas by identifying gaps in knowledge. Open and challenging issues in emerging smart park solutions are proposed for further work.

## 1. Introduction

The global population has been increasing and this is predicted to continue in the coming years. According to the United Nations report, the world population is expected to reach 8.6 billion in 2030, despite a slowdown in growth [1]. The same report also states that as of 2017, about 53.9% of the world’s population lived in cities, and by 2050, this number is projected to increase to 68.4%. Increased life expectancy is one of the factors contributing to aging and older populations, according to the World Health Organization (WHO) [2]. In developed nations, the percentage of the population over 60 years old is expected to increase from 20% to 33% between 2015 and 2050. This will nearly double the percentage of the world’s population over 60 years old from 12% to 22%.

Urbanization, a global trend characterized by the migration of people from rural areas to cities [3], has brought about significant changes in lifestyle, including an increase in sedentary behavior. As a result, urban settings face major public health challenges such as obesity and mental illness [4]. Research has established a link between low physical activity levels and age-related health issues, emphasizing the impact of urbanization on overall wellbeing (e.g., [5,6,7]). Moreover, an average of 36% of EU citizens aged 65 and older reported having at least two chronic diseases in 2020 [8]. Addressing these challenges requires a focus on population health and the recognition that urban areas not only encompass the built environment but also natural settings, urban green spaces, and public parks. These outdoor spaces have been shown to enhance individuals’ quality of life, physical and psychological wellbeing, and autonomy [4,9]. Furthermore, the concept of urbanization sparks ongoing debate as the boundaries between rural and urban areas become increasingly blurred and inaccessible for some. This recognition underscores the interdependence of urban areas, where components like cultivated fields, which contribute to the city’s sustenance, are considered to be part of the urban fabric. Without this interdependency and the scale and automation involved, these fields would not exist in their current form.

The potential benefits of smart public parks within the context of smart cities for older adults are widely acknowledged. However, there is a growing recognition that greater efforts are needed to ensure inclusivity and accessibility for this population [10]. Despite incorporating age-friendly design principles and safety-promoting technologies, the challenge of creating inclusive and accessible smart cities and smart public parks remains. This becomes a significant concern as the global older adult population continues to rise, emphasizing the importance of fully considering their needs and perspectives in the design and development of these technologies and spaces. To address this issue, it is crucial to advocate for the needs of older adults and actively involve them in the design and development processes of smart cities and smart public parks, ensuring that their perspectives and requirements are fully integrated and prioritized.

This article aims to provide a comprehensive overview of the latest research findings on smart public parks and emphasizes the significance of further exploration in this field. The paper makes several key contributions, including an extensive review of the current state of public park environments and their benefits for older adult users, an examination of technological advancements aimed at improving older adults’ health, an analysis of the existing development and technological solutions, and a discussion on the current challenges and unresolved issues pertaining to the research and implementation of smart public parks. The article is structured as follows: Section 2 delves into the definitions of health and wellbeing and links them to diseases related to urbanization; Section 3 explores the utilization and benefits of urban green spaces; Section 4 focuses on the concept of active aging in cities, encompassing topics such as smart cities, age-friendly cities, and smart parks; Section 5 highlights recent research on older users and technological solutions; and Section 6 addresses the challenges and open issues that emerged from this analysis. Finally, the article is concluded in Section 7.

## 2. Health and Wellbeing

In 1946, the WHO explicitly linked health to wellbeing by defining the former as “a state of complete physical, mental, and social wellbeing and not merely the absence of disease or infirmity” [11], whereas Brüssow defines health as “the capacity to adapt to changing external and internal circumstances,” thus making the concept of health broader [12]. In 2015, the WHO acknowledged that various social, economic, and environmental factors, individual behaviors, and medical interventions impact health, thus implying that any strategy for improving health and wellbeing should include the physical, mental, and social dimensions of people [2].

Wellbeing, encompassing individuals and societies, is a positive state influenced by social, economic, and environmental factors, representing a resource for daily life, quality of life, and the capacity to meaningfully contribute to the world [13]. According to Crisp, the term “wellbeing” is most frequently used in philosophy to refer to what is ultimately or non-instrumentally beneficial for an individual, and a person’s wellbeing is what is “beneficial” to them [14]. Therefore, while being healthy could be considered a component of wellbeing, it is not tenable to assume that it is the only factor. According to Keyes, wellbeing and illness are measured differently and are not mutually exclusive, highlighting that interventions can be implemented to increase wellbeing for individuals with diagnosed illnesses as well as those without but with low levels of wellbeing [15]. Moreover, the subjectivity of wellbeing is also an important aspect. According to Diener, E. et al., subjective wellbeing (SWB) refers to an individual’s own assessment and perception of how well their life is going, with the term “subjective” emphasizing the focus on personal evaluations and perspectives of life quality [16]. According to Diener and Ryan, while self-report measures are widely used in subjective wellbeing research and often demonstrate strong agreement across different measures, it is crucial to recognize the potential for measurement bias [17]. Hence, although subjective measures exhibit strong reliability when compared to non-subjective measures, the inclusion of non-self-report measures is valuable for offering a more comprehensive understanding of wellbeing and overall life satisfaction.

Physical health encompasses the overall wellbeing of the body, including its organ systems, immunity, and mobility [2,18]. The concept of physical health encompasses a wide range of outcomes, including subjective self-reports of symptoms and objective measures like mortality rates, with self-reported outcomes influenced by factors like memory biases, whereas objective disease endpoints provide more concrete and measurable indications of physical health [19]. Mental health, as defined by the WHO, is a state of wellbeing that enables individuals to effectively cope with life’s stresses, realize their abilities, learn, work, and contribute to their communities [20]. It goes beyond the absence of mental disorders and is an integral component of overall health and wellbeing. Mental health encompasses more than the absence of mental illness; it encompasses the presence of psychological wellbeing, which involves optimal psychological functioning and experience [21]. Social health pertains to an individual’s social wellbeing, including their ability to form and maintain relationships, engage in social activities, and feel a sense of connection to their community. Strong social connections provide emotional support, companionship, and a sense of belonging, making social health an essential aspect of overall health and wellbeing [22,23,24]. Keyes defined social wellbeing as how well a person perceives his or her relationships with others, neighbors, and community, and put forth five dimensions of social wellbeing that are theoretically supported: social integration, social contribution, social coherence, social actualization, and social acceptance [25].

According to Flies et al., these urban-associated diseases refer to those that either increase in prevalence or severity due to urban living or growth or are expected to rise as urbanization trends continue [26]. The study found evidence linking urban conditions to allergies and autoimmune, inflammatory, and infectious diseases, with air pollution being the most frequently associated health issue. Additionally, other urban risk factors such as altered microbial exposure, vitamin D deficiency, noise and light pollution, and the challenges of transient, overcrowded, and impoverished populations have been identified but are not as widely recognized.

Urban environments pose significant health risks and increase the prevalence of diseases among older individuals. Exposure to air pollution, a common feature of urban areas, has been linked to cardiovascular disease, which is a leading cause of mortality in older adults [27,28]. Furthermore, respiratory disorders like Chronic Obstructive Pulmonary Disease (COPD) and asthma are more prevalent among older individuals residing in urban settings, primarily due to the detrimental effects of air pollution [29,30,31]. The risk of developing type 2 diabetes is also heightened in urban areas due to factors such as chronic stress, unhealthy dietary choices, and sedentary lifestyles [32,33]. Urban living can contribute to higher rates of depression and anxiety in older individuals because of social isolation, limited social support, and exposure to crime and violence [34,35,36]. Sedentary behavior, which is common in urban lifestyles, can lead to weight gain and obesity, further increasing the risk of various health issues including heart disease, diabetes, and certain types of cancer [37,38]. Moreover, a lack of exercise can exacerbate osteoporosis, a condition characterized by reduced bone density and an increased susceptibility to fractures, emphasizing the importance of physical activity in maintaining bone health among older adults [39,40]. In the context of active aging, it is crucial to recognize that policies and programs supporting social connections and mental wellbeing are equally essential alongside initiatives focused on enhancing physical health [41].

## 3. Urban Green Spaces and Benefits

The concept of an urban green space (UGS) does not have a single, consensual definition [42]. The industrial revolution’s massive urbanization led to the development of the UGS concept in the 19th century [43,44,45]. The decline in natural landscapes within cities due to urbanization during the 20th century raised public awareness of the need to incorporate natural assets and components into urban contexts. This led to the development of the urban park movement, which was started to enhance urban life [46,47]. Industrialization was more noticeable in Europe and North America, which led to a greater emphasis on UGS. For instance, the idea of a “Green Lung” to purify city air was first implemented in Central Park, a 19th-century structure in North America. In addition, Fredrick Law Olmsted developed a brand-new idea of continuous green spaces in 1867 and named it a parkway [48,49].

UGSs, including forests, public parks, and community gardens, are intended to offer a variety of opportunities for any resident to interact with nature and partake in activities like exercise, relaxation, or socializing [50,51]. Additionally, they are essential for cities because they offer a variety of recreational opportunities, encourage social interaction and integration, and enhance mental and physical health [52,53]. Although parks are essential, most UGS planning thus far has been seen as being closely related to urban and garden design, rather than as a matter of public health [46]. The social role of UGS is, therefore, typically highlighted in relation to two main sets of concerns: first, those related to the practice of physical activity and relaxation, and second, those related to enhancing social and intergenerational cohesion [54].

UGSs may not only make cities more pleasant to live in but also serve social, cultural, aesthetic, practical, economical, and ecological purposes [54,55]. Consequently, the importance of UGS is closely related to a person’s level of self-care. Body mass index, subjective health assessments, and longevity can benefit from physical activity, relaxation, and good mental health [54].

Studies have shown that green spaces can positively affect older adults’ health. For instance, older people who lived close to green spaces in the Netherlands reported better health than those who did not [54]. Additionally, it has been discovered that green spaces can strengthen social ties and a sense of community while encouraging physical activity in seniors aged 60 and older [42]. Older people need to interact with others to be healthy and happy, and social isolation has been linked with higher mortality rates [56].

It is vital to consider the advantages of vigorous and moderate physical activity when discussing the advantages of green spaces for older adults and the general public. Studies from different countries have demonstrated that having access to and using green spaces can increase physical activity, reduce sedentary time, and promote leisure walking [42].

The benefits associated with UGS can only be fully realized when individuals have the necessary resources and time, as well as a balanced quality of life. This includes factors such as having the opportunity to engage in UGS activities and achieving a satisfactory work–rest balance. One crucial aspect that enables the realization of these benefits is the presence of a proper retirement system. With a well-planned retirement, individuals can take advantage of the benefits of UGSs and enjoy a higher quality of life.

Green spaces in neighborhoods are essential for long, leisurely walks [57]. Therefore, it is essential to develop green spaces that motivate seniors to engage in moderate physical activity if you want to advance public health. This is crucial because many older people struggle to maintain moderate physical activity levels [42]. Physical activity contributes to overall wellbeing, cardiovascular health, mental health, neurocognitive growth, and the prevention of obesity, cancer, and osteoporosis [58]. Inactivity is one of the primary risk factors for mortality worldwide [59].

Moreover, inactivity affects the global population’s overall health and the prevalence of non-communicable diseases [60]. Research has shown that green spaces and physical activity levels are related [61]. Physical activity in green spaces is particularly beneficial for urban dwellers with a mental illness [62]. Other demographics or subgroups may also experience similar benefits from green space, which makes outdoor activities enjoyable and convenient and promotes less sedentary lives. An analysis of ten studies conducted in the United Kingdom demonstrated multiple mental health benefits from physical activity in green environments [63]. ‘Green exercise,’ which may be defined as a physical activity undertaken in green or natural environments, has been suggested to be more beneficial than other types of exercise [63,64].

Additionally, physical activity is linked to improvements across mental health indicators [65,66]. A lack of exercise can exacerbate health issues, especially in older adults [67,68]. Some diseases get worse with age [69]. Moreover, it is evident that physical activity plays a significant part in active aging [70]. However, according to [71], active aging encompasses more than just physical activity in older individuals, incorporating individual, social, and physical aspects, as well as the policy-making process and environmental factors that influence physical activity, health, and the overall context in which these activities take place. Age-friendly cities can provide a safe environment for older adults to leave their homes and participate in a society that values older adults and becomes more engaged with them.

## 4. Active Aging and Cities

The aging process refers to the biological changes that occur over time, resulting in a gradual loss of physiological integrity, diminished function, and increased mortality risk [72]. Aging involves the deterioration of bodily functions and a decline in physical and mental capacity, primarily driven by cellular damage [72]. While aging is a primary risk factor for various diseases, including cancer, diabetes, cardiovascular disorders, and neurodegenerative diseases, it is important to recognize that aging itself is not a disease but a natural phenomenon [41,72,73]. Rather than focusing solely on disease treatments, adopting health-oriented strategies becomes crucial in addressing the impacts of aging. While there is no cure for aging, promoting active aging can play a significant role in maintaining overall wellbeing.

There is no widely accepted concrete definition of active aging. The concept of ‘Active Aging’ was initially introduced by Kalache who established a correlation between engagement in activities and the promotion of health in later life [74]. From Kalache’s perspective, the essentiality of providing older individuals with ample opportunities to maintain an active lifestyle is emphasized, as good health, acting as a catalyst for individual and societal contributions, is dependent on both personal efforts and societal support. Consequently, sustaining activity and embracing an active life significantly enhance the likelihood of attaining optimal health in older age [74]. An active life means the best chance of being healthy. According to the WHO, active aging is the process of maximizing opportunities for health, participation, and security to improve quality of life as people age [41]. The European Commission characterizes active aging as assisting people in continuing to live independently as they age and, whenever possible, contributing to the economy and society [75].

According to the WHO, active aging is the process of maximizing opportunities for health, participation, and security to improve quality of life as people age [41]. It is linked to various life transitions and involves maintaining health through activities that align with individuals’ goals, capacities, and community opportunities [71]. Healthy habits, such as a balanced diet and regular exercise, are emphasized as important components of active aging, reducing the risk of diseases, and enhancing physical and mental wellbeing. Determinants of active aging include economic, health, social service, behavioral, individual, physical environment, social, cultural, and gender factors, highlighting the need for localized studies and information gathering to develop effective strategies for older adults. It is worth noting that there is a divergence between European and US approaches toward active aging, with Europe prioritizing health and wellbeing, while the US focuses more on productivity [76].

Promoting healthy aging and ensuring a high quality of life in an aging population are significant concerns in society, emphasizing the importance of maintaining wellbeing and healthy aging. Personal traits and environments play a crucial role in determining healthy aging, with research highlighting their greater influence compared to external factors [77]. Studies indicate that health is influenced by physical and social environments, as well as rewards and obstacles that affect opportunities, decisions, and health behavior (e.g., [78,79,80,81]). Multiple dimensions, including physical, cognitive, and social factors, are considered in defining healthy aging. Hansen-Kyle defined healthy aging as a process that involves slowing down physically and cognitively while resiliently adapting and compensating to optimize functioning and participation in all areas of life, including physical, cognitive, social, and spiritual aspects [82].

In addition, it is essential to identify and eliminate barriers that hinder older adults from engaging in the community and ensure that their voices are heard [83]. The same study notes that in 2020, the number of adults aged 60 and above exceeded the population of children under the age of five. Therefore, as the population ages, it is crucial to prioritize age-friendly cities that enable older adults to maintain an active lifestyle. By implementing supportive policies, services, and infrastructure, age-friendly cities can ensure that older adults have access to social participation, healthcare, transportation, and other essential resources that promote an active and inclusive lifestyle. From smart homes and assistive technologies to digital healthcare systems and transportation advancements, integrating technology into age-friendly city planning can create innovative solutions that empower older adults and enable them to thrive in their communities.

### 4.1. Age-Friendly Cities

Older residents are a valuable resource, but our cities must ensure their inclusion and full access to urban spaces, structures, and services in order to fully realize their potential for continued human development [84]. A comprehensive guide by the WHO outlined criteria for cities to be classified as “age-friendly”, drawing upon the WHO’s framework for active aging [77]. The guide focused on eight key areas, including housing, outdoor spaces and buildings, social participation, respect and social inclusion, civic engagement, employment, communication and information, and community support and health services.

Our cities must prioritize including older residents and giving them full access to urban spaces, structures, and services in order to fully realize the potential for continued human development among older residents [84]. The comprehensive guide provided by the WHO offers valuable suggestions and criteria for cities to become “age-friendly,” drawing upon the WHO’s framework for active aging [77]. This guide focuses on key areas such as housing, outdoor spaces and buildings, social participation, respect and social inclusion, civic engagement, employment, communication and information, and community support and health services, serving as a valuable resource for cities to enhance their age-friendliness and ensure the wellbeing of older residents. A further suggestion made by Dash et al. is the creation of an ecosystem that is age-friendly and includes cities, communities, health systems, universities, and employers [85].

As highlighted by the WHO document, government policies have been shaped by the analysis and expression of the older adult population’s circumstances, leading to their active involvement in decision making [77]. The endorsement of this approach by the United Nations in 2007 signifies the recognition of older individuals’ ability to contribute to society. Key factors such as outdoor spaces and buildings, transportation, and housing, which are closely linked to personal mobility, safety, health behavior, and social participation, are considered to have the most significant impact on an age-friendly city, according to the WHO [77]. Moreover, an age-friendly city is characterized by policies, services, settings, and structures that support active aging, acknowledge the diverse capacities and resources among older people, cater to their changing needs and preferences, respect their choices and lifestyles, protect the vulnerable, and promote their inclusion and contributions to all aspects of community life [86].

According to the WHO, the presence of safe and accessible public buildings, transportation systems, and pedestrian-friendly spaces exemplifies supportive environments that can enhance the preparedness of cities [86]. Age-friendly cities are not only designed to cater to people of all age groups but they have a particular focus on the older population. These cities encompass policies, services, and infrastructure that foster healthy and active aging, empowering older individuals to contribute to society and ensuring that they can live with dignity, security, and enjoyment. Key features of age-friendly cities often include the accessibility of facilities for senior citizens and their active participation as valued members of the community (e.g., [87,88]).

In their study, Rashid et al. examined various reports from age-friendly initiatives around the world, including those from New York City, Taiwan, Washington DC, London, Liverpool, South Australia, the UK, and New Zealand [89]. By applying the eight dimensions outlined in the WHO’s guide, the authors identified 60 age-friendly features and categorized them according to the most relevant dimensions. These findings underscore the importance of considering the specific needs of older adults when developing city policies, services, and infrastructure to foster inclusivity and wellbeing for all residents.

The study narrowed down Rashid’s original eight dimensions to seven, leaving out dimension three, which is associated with housing features. Each of the seven dimensions also excluded certain characteristics that were not directly relevant to smart parks. The seven revised dimensions, resulting in 33 features, are shown in Table 1.

One piece of research conducted analyses on multiple reports at different levels of governance, including district, city, state, and federal governments [89]. This variability highlights that the WHO guidelines provide principles that can be adapted and applied across various levels and regions [77]. It is important to categorize the key features of different elements within a city. For instance, buildings and outdoor spaces play a crucial role in public parks. By understanding the specific features that are important for different city elements, urban planners and policy-makers can take more effective actions to create age-friendly environments.

In 2018, the WHO introduced a report on “The Global Network for Age-friendly Cities and Communities,” which proposed a continuous improvement cycle for creating age-friendly environments [86]. This cycle consists of four dimensions: Engage and Understand, Plan, Act, and Measure, as depicted in Figure 1. Starting with the “Engage and Understand” dimension, the process progresses through the other dimensions until reaching the “Measure” dimension. This cycle is not a closed loop and can be initiated again from the “Engage and Understand” dimension. The age-friendly environment continuous cycle shares similarities with the Design Thinking method, which typically follows a five-step model: Empathize, Define, Ideate, Prototype, and Test [90]. Both strategies prioritize user-centered approaches, emphasizing the value of co-design and participatory design in creating effective and user-friendly solutions.

Ref. [87] claims that creating age-friendly communities has grown into a significant social policy concern, encompassing issues relating to both urban and rural settings. As cities globally embrace the smart city concept, which harnesses information and communication technologies (ICTs) and Internet of Things (IoT) solutions to enhance the wellbeing of citizens and address urban resource management challenges, it becomes increasingly crucial for urban development to prioritize the formulation of new policies and strategies aimed at integrating older adults into the social and economic fabric of cities [87,91]. Age-friendly cities focus on creating environments that cater to the needs of older adults, ensuring accessibility, safety, and social engagement, while smart cities leverage technology to enhance various aspects of urban life, such as transportation, energy efficiency, and infrastructure management. By integrating age-friendly principles with smart city solutions, cities can create innovative and inclusive environments that prioritize the wellbeing of all residents, regardless of age.

### 4.2. Age-Friendly and Smart Cities

Modern technology has radically changed how we think about cities. The concept of a “Smart City” was first introduced in the book “The Technopolis Phenomenon: Smart Cities, Fast Systems, Global Networks,” which marked the beginning of the study of “Technological Cities” in the early 1990s [92]. Nevertheless, the idea of a smart city was not studied until the late 2000s. A smart city can be characterized in many ways.

Different definitions of smart cities are available nowadays; bodies such as the European Commission, the Institute of Electrical and Electronics Engineers (IEEE), and the United Nations have different views on what constitutes a smart city. According to the European Commission, smart cities are places where technological innovations are applied to improve urban management and productivity, whereas the European Commission defines a “smart city” as a location where traditional networks and services are improved to benefit its citizens and businesses using digital and telecommunication technologies [93]. For the European Commission, “A smart city goes beyond the use of Information and Communication Technologies (ICT) for better resource use and fewer emissions. It means smarter urban transport networks, upgraded water supply and waste disposal facilities, and more efficient ways to light and heat buildings. It also means a more interactive and responsive city administration, safer public spaces, and meeting the needs of an aging population”.

Hammons and Myers have a more technological approach to the smart city concept, defining it as a place that “brings together technology, government, and society and includes but is not limited to the following elements: a smart economy, smart energy, smart mobility, a smart environment, smart living, and smart governance” [94]. The United Nations characterizes smart cities through the definition of the International Telecommunication Union (ITU).

The ITU defines a smart city as “an innovative city that uses ICTs and other means to improve quality of life, the efficiency of urban operation and services, and competitiveness while ensuring that it meets the needs of present and future generations concerning economic, social, environmental, as well as cultural aspects” [95].

According to Patrão et al., a smart city is a well-balanced blending of human, social, cultural, economic, environmental, and technological developments integrated to overcome the main challenges of urban living [96]. The authors affirm that technology should be viewed as a means of implementing a smart city following the requirements of its location (environment, energy, people, business, and governance). Another crucial component of a smart city, according to Lnenicka et al., is its capacity to enable and empower citizens and support individual and collective demands for wellbeing by fusing intelligent technologies with the built and natural environments [91]. This strategy is based on what technology can offer to citizens rather than just what it can do for planners or city managers. According to the authors, a smart city uses IoT and ICT solutions in everyday life to improve the quality of life of its residents and help local governments to address issues with the use, reallocation, and provision of urban resources. ICT can support transparency, ensure that decision-makers are held accountable, and encourage citizen participation in governance. Moreover, Fernandes claims that networked services will enable the development of city management systems that effectively use resources, assist in reducing potential damage, and make cities resilient to be able to handle potentially dangerous situations flexibly [97].

In summary, the IoT and ICT underpin most conceptualizations of smart cities, serving as the main drivers in enhancing citizens’ quality of life and facilitating decision making.

Smart cities and age-friendly cities share some common features, including the following:A focus on technology: In smart cities, technology aims to improve efficiency, sustainability, and economic development. In age-friendly cities, technology supports healthy aging and provides access to services and opportunities.Livability: Enhancing the physical environment to make the community more habitable for residents. Providing access to services and amenities. Creating opportunities for social engagement.Accessibility and inclusion: ensuring that all residents, especially those with disabilities, older adults, and other marginalized groups, can participate in the community.Collaborative approach: facilitating collaboration and partnerships between government, community organizations, and the private sector to achieve their goals.Data-driven decision making: relying on data from various sources, such as sensors and surveys, to make informed decisions and track progress toward their goals.

Even though the primary goals of age-friendly cities and smart cities differ in some ways, both can improve the quality of life for most residents by addressing their needs. Both can enhance key city components like services, transportation, health, or public areas.

As noted earlier, it is clear that the concept of smart cities is far from consensual (and thus objective). Moreover, each definition of a smart city carries its own epistemic and political compromises, which means that some definitions will focus on technocratic efficiency, whereas others might lean toward a more sociotechnical view. Moreover, since smart cities require a vast amount of data, it is important to acknowledge the potential dangers that the IoT might bring in terms of privacy. Privacy is in itself a contested issue, as its very definition is open for debate [98]. Consequently, Western European conceptualizations of privacy are not necessarily the same as those in Eastern countries [99]. This is not to say that privacy should be ignored, nor that certain implementations of smart cities—particularly in China—might be worrisome [100]. Nonetheless, in this paper, the idea of a smart city is a broader concept that serves as a scaffold for discussing the role of public parks and how they might integrate this concept; therefore, deeper discussions about the ethics of smart cities are beyond this paper’s scope.

### 4.3. Smart Public Parks

Public parks play a vital role in cities as they comprise various elements like streets, buildings, and open spaces, necessitating attention for the development of smart cities’ “2.0 version” of smart parks. The concept of integrating smart devices and technologies in public parks holds significant potential, transforming them into smart parks that provide a multitude of benefits and services. However, there is a need for a clearer understanding of the exact definition and features of smart parks, despite their promising prospects.

Lele and Lihua propose that smart parks aim to transform interactions between the government, enterprises, and residents by offering abundant smart services and enabling intelligent park operations [101]. They identify three key features for smart parks: perception, interconnection, and intelligence. Perception involves accurately monitoring critical objects using IoT technologies, interconnection establishes networks to connect park systems and departments, and intelligence focuses on autonomous management systems with data integration and analysis for informed decision making.

While the majority of smart services in domestic and overseas parks have primarily focused on urban efficiency, such as safety, crime prevention, and environmental maintenance, Lee argues that parks are natural spaces that require services aimed at nature, human wellbeing, and community recovery [102]. Therefore, the application of technology in smart parks should prioritize restoring these elements rather than merely showcasing the technology itself. For the effective planning and implementation of smart park services, Lee suggests close collaboration between park planners and IT experts. Additionally, careful consideration and evaluation of costs, effectiveness, user satisfaction, and sustainability are necessary.

Loukaitou-Sideris et al. define smart parks as parks that incorporate various technological innovations, including environmental, digital, and material technologies, to achieve multiple values such as equitable access, community fit, enhanced health, safety, resilience, and efficient operations and maintenance [103]. These parks aim to seamlessly integrate into their socio-physical surroundings, ensure easy accessibility, promote community health and safety, and demonstrate resilience to climate change, while being energy-efficient. By leveraging technological advancements, smart parks can enhance their performance, reduce long-term costs, and provide an array of benefits to the surrounding communities.

Lee et al. highlight the need for public communication and increased user awareness when implementing smart technologies in parks [104]. While the focus has often been on safety and environmental areas, the lack of public engagement and the nature of maintenance-oriented services have resulted in low awareness among park users.

Furthermore, Kim et al. emphasize that smart parks should not only support citizens’ safe and enjoyable use of parks but also enhance management and operational effectiveness [105]. By utilizing digital, environmental, and material technologies, smart parks contribute to the social, economic, and environmental sustainability of cities and local communities.

It is important to note that smart parks are still in the early stages of development and are considered to be a crucial component of strategic planning for smart cities. In conclusion, the development of smart parks represents a significant opportunity for cities in their journey toward becoming smart cities. By integrating smart devices and technologies, public parks can offer a multitude of benefits and services to enhance urban efficiency, promote community wellbeing, and foster environmental sustainability. The concept of smart parks encompasses key features such as perception, interconnection, and intelligence, aiming to transform park operations and provide abundant smart services. However, further research and collaboration between park planners, IT experts, and the community are essential in ensuring the effective planning, implementation, and user awareness of smart park initiatives. With ongoing advancements in technology and a focus on restoring nature, enhancing user experiences, and achieving sustainability goals, smart parks have the potential to optimize benefits for individuals, communities, and the surrounding environment within the broader context of smart city development.

In the forthcoming section, an analysis of technologies and solutions is provided specifically targeted at older adults. This analysis will encompass a comprehensive examination of various technological advancements and innovative solutions that cater to the unique needs and challenges faced by the older population. Through this analysis, we hope to identify key trends, opportunities, and potential areas for further research and development in the field of technology for older adults, ultimately contributing to the advancement of age-friendly and inclusive solutions.

## 5. Recent Research on Older Adults and Technological Solutions

This section presents an analysis and discussion of existing research on smart public parks, exploring their impact on promoting healthier aging among older adults. It also highlights solutions and recommendations for the design or redesign of parks with a focus on senior citizens. Extensive studies have been conducted on the relationship between public parks and the wellbeing of older individuals. To provide insights into the current state of research, a comprehensive analysis of 28 recent articles was undertaken, categorizing them based on their objectives, environmental aspects, proposed solutions, research methodologies, and geographical locations (see Table 2).

A search of the Elicit database was conducted using the phrase “Smart parks for elderly wellbeing,” with a focus on the most widely cited papers released between 2020 and 2023. The search requested abstract summaries, main findings, study types, regions, and population summaries from the original research. A total of 31 articles related to the topic were identified by Elicit. These articles encompass a diverse range of approaches. However, three articles were excluded: one was an earlier version of research by the same author, the second lacked results as it focused on research methodology, and the third article was unavailable. The approaches found in the remaining 28 articles varied from technological solutions to a more social perspective of the problem. It is worth mentioning that Elicit states its limitation to publications in Semantic Scholar, excluding licensed journals and articles behind paywalls, which may result in a gap in the literature being retrieved [106].

The majority of the papers analyzed in this study originate from Europe (n = 15) and China (n = 5), indicating the geographic distribution of research in this field. Notably, a diverse range of solutions encompassing policy-related, design-related, technological-related, and mixed approaches were identified. Figure 2 illustrates these four types of solutions. Among the different types of research conducted, reviews (n = 10), qualitative studies (n = 7), and applied studies (n = 7) were the most prominent categories, while data analysis studies or quantitative research (n = 4) represented a smaller portion. Data analysis studies primarily rely on open-access data and often utilize Geographic Information System (GIS) research methodologies, such as the visitation pattern analysis described in [52]. Qualitative studies employ questionnaires, observations, and interviews to gain insights into the specific challenges faced by park users in particular regions. For instance, Onose et al. focused on understanding the interactions, needs, and motivations of older individuals in Romanian public parks [107]. Literature reviews aim to establish parameters and guidelines for future research. As an example, Alves et al. defined parameters for constructing a walkability index map based on non-GIS information [108]. Lastly, applied studies concentrate on solutions, predominantly relying on wearables. An example is Lachtar et al.’s work, where they developed a cane utilizing LoRa and MQTT technologies for tracking older adults [109].

Furthermore, the analysis revealed that the articles examined in this study encompassed both indoor and outdoor research and solutions. Indoor studies were predominantly centered around enhancing elderly autonomy and providing healthcare assistance. On the other hand, outdoor research focused on evaluating the suitability of cities for older adults, such as by assessing walkability indices and visitation patterns, and addressing issues related to public parks. Notably, a greater number of studies emphasized the use of smart devices in outdoor environments. This prevalence of outdoor solutions in the applied research findings may be attributed to the specific keywords used to retrieve the articles, highlighting the importance of keyword selection in shaping research outcomes.

**Table 2 geriatrics-08-00094-t002:** Categorization of articles.

Authors	Objective	Environment	Solution	Type of Research	Country
[52]	Overview of visitation patterns	Outdoor	Better planning	Data analysis	Germany
[110]	Understand the impact of built environments in older adults’ physical activity	Outdoor	Design solution	Review	Italy
[111]	Smart environments and social robots for age-friendly integrated care services	indoor	Technological	Review	Romania
[112]	Smart residential environments for older adults	Indoor	Technological	Review	South Korea
[109]	Development of a cane with LoRa and MQTT	Outdoor	Technological	Applied research	Tunisia and France
[113]	Wearable biosensors and hotspot analysis used to detect stress areas	Outdoor	Technological	Applied research	United States of America
[107]	Understand how older adults interact inside public parks	Outdoor	Better planning	Qualitative	Romania
[114]	Understand why older adults visit public parks	Outdoor	Better planning	Qualitative	Australia
[115]	Understand the factors that influence older adults’ psychological wellbeing	Outdoor	Better planning	Qualitative	Taiwan
[116]	Understand the impact of multifunctioning in public parks	Outdoor	Better planning	Qualitative	Sweden
[117]	Health-related ICT solutions to smart environments for older adults	Indoor	Technological	Review	Czech Republic, Bosnia and Herzegovina, and North Macedonia
[108]	Development of a walkability index for older adults’ health	Outdoor	Mixed solution	Review	Portugal
[118]	Development of a tool to support decision makers in the development of policies aimed at improving pedestrian accessibility to urban services	Outdoor	Mixed solution	Applied research	Italy and United Kingdom
[119]	Analysis of institutional and individual conditions for a new concept of the smart development of ageing communities	Outdoor	Better planning	Data analysis	Poland
[120]	Seniors’ physical activity in neighborhood parks and park design characteristics	Outdoor	Better planning	Data analysis	China
[121]	Innovative and assistive eHealth technologies for smart therapeutic and rehabilitation outdoor spaces for the older adult demographic	Outdoor	Technological	Review	New Zealand and Australia
[122]	Urban park facility use and intensity of seniors’ physical activity	Outdoor	Technological	Data analysis	China
[123]	Effects of the built and social features of urban greenways on the outdoor activity of older adults	Outdoor	Better planning	Qualitative	Taiwan
[124]	Smart bus stops as interconnected public spaces for increasing social inclusiveness and quality of life of elder users	Outdoor	Technological	Applied research	Spain
[125]	Smart nursing homes: self-management architecture based on the IoT and machine learning for rural areas	Indoor	Technological	Applied research	Spain
[126]	Healing spaces: improving health and wellbeing for older adults through therapeutic landscape design	Outdoor	Design solution	Review	New Zealand
[127]	Impact of a low-cost urban green space intervention on wellbeing behaviors in older adults: A natural experimental study	Outdoor	Design solution	Qualitative	United Kingdom
[128]	Smart cities’ applications to facilitate the mobility of older adults	Outdoor	Technological	Review	Portugal
[129]	An IoT-enabled smart living environment for older adults	Indoor	Technological	Applied research	United States of America
[130]	The concept of smart city in terms of improving the quality and accessibility of urban spaces for older adults	Outdoor	Technological	Review	Poland
[131]	Suitability of park recreational space layout for older adults based on visual landscape evaluation	Outdoor	Mixed solution	Qualitative	China
[132]	Data-driven winter landscape design and pleasant factor analysis of older adult friendly parks in severe cold cities in Northeast China	Outdoor	Technological	Applied research	China
[133]	Research on older adult friendly parks based on inclusive design concept	Outdoor	Design solution	Review	China and South Korea

The review highlighted several factors and barriers that influence the motivation of older adults when considering a visit to a public park. These motivations were categorized into three main activities within the park: engaging in physical activity, participating in visitation and relaxing activities, and seeking social interactions. By understanding these factors and barriers, it becomes possible to better comprehend the drivers behind older individuals’ decision-making processes and their preferences for different park activities.

In terms of physical activity, Bonaccorsi et al. identified positive factors such as walkability, urbanization, street connectivity, access to facilities, pedestrian-friendly infrastructure, green spaces, safety measures, and aesthetic qualities that motivate individuals to engage in physical activity [110]. Conversely, negative factors such as barriers to walking, poor infrastructure, safety concerns, pollution, and noise discourage physical activity. Additionally, Veitch et al. highlighted the importance of walking paths, organized activities, and fitness equipment in promoting physical activity among seniors [114]. The presence of outdoor fitness equipment has been found to positively impact the total steps and energy expenditure of older adults, offering alternative exercise options beyond walking [120]. Living in areas with abundant amenities and resources is associated with leading more active lifestyles and increasing walking time [134]. Furthermore, proximity to greenways with higher levels of neighborhood social capital and well-maintained paths, natural elements, and seating is linked to increased outdoor activities and subsequent physical activities among residents [123].

According to Stanley et al., older users prefer a higher number of benches in public parks as they are inclined toward relaxation and enjoying the scenery [135]. Therefore, benches can serve as motivators for their park visits. Enssle and Kabisch found that older adults with stronger social networks take more visits to parks compared to those who are socially isolated [52]. The study also identified trees, nature and greenery, walking paths, and seating as crucial park features desired by visitors, particularly older adults. Furthermore, attractive, peaceful, and relaxing park environments are highly favored by senior citizens. Veitch et al. confirmed that well-maintained parks with established trees, gardens, birdlife, seating, pleasant paths, toilets, cafés, water features, shade, facilities for grandchildren, and the presence of other people encourage older adults’ visitation [114]. The authors also highlighted that picnic/barbecue areas, scheduled events, cafés, and aesthetically pleasing surroundings act as motivators for social interaction among senior park visitors.

The review categorizes into three different types of solutions (design, management, and technological), as illustrated in Figure 2. The analysis indicates a strong emphasis on technological solutions (n = 13) for enhancing the wellbeing of older adults, reflecting the prevalence of technology-based approaches. Additionally, improved planning, policy, and management solutions were identified as being crucial (n = 7), underscoring the significance of advancements in these areas. On the other hand, design solutions (n = 3) and hybrid solutions (n = 3) have received less attention, indicating opportunities for further exploration and integration. These findings shed light on the existing research gap in terms of integrated solutions and highlight the potential for hybrid approaches to address the complex needs of optimizing quality of life for older adults in public park settings.

Exploring public park issues involves various techniques and offers diverse solutions, including technological, policy-related, design-related, or hybrid approaches. Table 2 presents two distinct methods taken in different studies. The first method focuses on identifying issues and understanding the population’s needs. These studies (e.g., [107]) delve into user motivations, perceived problems, and user studies conducted in specific geographic areas like cities, nations, or groups of nations/cities. They employ techniques such as observations, interviews, questionnaires, or literature reviews to comprehensively grasp why people visit public parks. This method aims to provide practical and realistic solutions by building upon the existing knowledge of the issues and population (e.g., [109]). While all of the studies in Table 2 offer expert recommendations or solutions, none of them follow a systematic approach. Table 2 also highlights several recommendations from the articles. The active participation of older users in the planning process is crucial in order to design age-friendly parks [52]. Citizen participation ensures better outcomes and addresses the specific needs of the older population.

Researchers must develop an evaluation framework or principles guide that can be universally used to assess the emotional needs and technology engagement of older adults in order to advance the field’s research beyond individual studies [113]. Furthermore, conducting studies on intelligent environments that promote active behaviors among older adults can enhance our understanding of the interplay between technology and design, bridging knowledge gaps across disciplines. When applying research findings from one country to another, caution should be exercised, as these findings may be influenced by cultural or geographical factors [110]. It is recommended that qualitative techniques, such as focus groups, interviews, and observations, be combined with objective measurements like comfort and usability to obtain more precise data on the quality of bench designs. The concept of the Uncertain Geographic Context Problem (UGCoP) highlighted by Kwan refers to situations where contextual variables and research findings are sensitive to different delineations of contextual units [136].

It is critical for policy- and decision-makers, urban planners, landscape architects, and government organizations at various levels to comprehend the reasons why older people visit public parks to ensure that parks effectively support older adults’ health and active lifestyles [114]. This highlights the importance of developing policies that specifically cater to the needs of senior park users. Furthermore, promoting user participation in park management and creating socially multifunctional spaces can help to create inclusive parks that cater to diverse age groups [116]. It is important for planning policies and practices to be more nuanced and tailored to the specific requirements of older adults [127]. Landscape architects can contribute by incorporating outdoor exercise equipment in urban parks, encouraging seniors to engage in physical activity [120]. Understanding the reasons, approaches, and optimal timing for senior citizens to use outdoor exercise equipment is essential for designing effective interventions.

As stressed by Podgórniak-Krzykacz et al., it is crucial to incorporate a citizen-centered approach to create smart and age-friendly environments [119]. Similarly, Tian et al. highlight the importance of human-centered design that promotes fairness, flexibility, diversity, and positive user experiences, while considering the specific space usage needs of older adults [133]. Marques et al. recommend a person-centered strategy that focuses on usability and accessibility, gathering insightful data, and offering feedback to address potential resistance to technology adoption to ensure the successful adoption of eHealth technologies [121]. Furthermore, Marques et al. emphasize the importance of accuracy, security, and reliability in addition to user-friendliness, ubiquity, and user-centricity when developing technologies for older adults.

Onose et al. point out the significance of connecting public parks to the nearby neighborhood to maximize their use [107]. In terms of park facilities, Zhai et al. suggest that seniors prefer amenities such as flat pavements and benches [122]. Creating engaging environments with features like artwork, sculptures, and cafes is crucial, as emphasized by Enssle and Kabisch and Veitch et al., as these elements can attract older adults [52,114]. Additionally, providing appropriate fitness equipment can encourage more active use of the parks.

Chang et al. suggest that urban greenways, linear public spaces next to roads, should prioritize environmental quality to create outdoor spaces that promote the wellbeing of older adults and are simple to maintain [123]. Additionally, Marques et al. suggest recommendations for smart therapeutic landscapes, including the use of on-site and remote monitoring tools, activation mechanisms for assistive technologies during emergencies, and the localization of affected senior citizens [126]. These measures aim to ensure that older adults can enjoy the outdoors while addressing their specific needs and concerns.

## 6. Challenges

The global population is undergoing an inevitable process of aging, as highlighted by Tian et al. [133]. However, society should view this aging population as an opportunity to successfully adapt to this new demographic reality, as emphasized by Maresova et al. [117]. Public parks and urban green spaces offer older individuals venues for entertainment and social interaction, and opportunities to stay physically active.

Specific elements of the built environment that facilitate physical activity among older adults were identified by Bonaccorsi et al. [110]. These include factors such as walkability, residential density/urbanization, street connectivity, land-use mix-destination diversity, overall access to facilities, destinations, services, pedestrian-friendly infrastructure, greenery, aesthetically pleasing scenery, high environmental quality, street lighting, crime-related safety, and traffic-related safety. To fulfill the design requirements and promote the inclusive planning, design, and management of green spaces, it is crucial to consider attractive features and address park-related issues, as emphasized by Onose et al. [107]. In this regard, ICT can play a valuable role in facilitating the monitoring process.

Cities must prioritize inclusivity and accessibility for all citizens, regardless of age or physical condition, in order to be deemed age-friendly. According to Onose et al., it is crucial that public parks are designed with the needs, demands, and preferences of all visitors in mind [107]. Creating inclusive communities that welcome residents of all ages requires collaboration among various stakeholders and active citizen involvement [119]. A supportive political climate is also essential to foster innovation and drive positive local changes.

Rather than solely focusing on technologies that extend lifespans, it is important to consider intelligent services and technologies that enhance the wellbeing of older adults in their living environments [112]. Podgórniak-Krzykacz et al. argued that efforts should be made to educate older adults on digital skills and implement technological advancements in public spaces to enable them to live well [119]. The IoT can play a significant role in monitoring health conditions and providing enhanced security for older individuals, such as through emergency response mechanisms, fall detection solutions, and video surveillance systems. Additionally, Marques et al. noted the increasing popularity of senior fitness areas in public spaces [121]. Planners should prioritize the inclusion of outdoor fitness equipment and natural areas, particularly in regions where obesity and physical inactivity are prevalent [120]. Smart public parks can be powerful tools in addressing these challenges with remarkable efficacy. Leveraging their inherent intelligence and advanced capabilities, these technologically enhanced parks have the potential to accurately identify and swiftly respond to pressing issues. By integrating smart technologies and harnessing data-driven insights, these parks can proactively tackle concerns such as safety, maintenance, and visitor experience, ushering in a new era of optimized park management and ensuring an enjoyable and sustainable environment for park visitors of all ages.

Ensuring that technological solutions effectively meet the needs of users, particularly older adults, is crucial [128]. Personalization is key to enabling older users to adapt technology to their specific requirements. However, several studies have highlighted the insufficient attention given to important issues such as user privacy, data standardization and integration, IoT implementation, and sensor characteristics [137,138,139]. Moreover, research has shown that IoT devices designed for older adults often lack personalization and fail to adequately consider their needs and preferences [111]. Therefore, customization and personalization should be given top priority in technological solutions to improve the user experience for older adults. However, important issues like user privacy, data standardization, and IoT implementation must be addressed. Adopting a discreet and unobtrusive approach, rather than solely relying on older adults, can ensure the effectiveness and acceptance of technological solutions for this demographic.

There is a significant misalignment between the current state of knowledge and the marketing of research and development (R&D) outcomes, with most solutions still in the early stages of prototyping and development [117]. Home automation has made notable advancements in monitoring remote health. Studies (e.g., [109,112,113,129]) have shown that IoT technologies can play a crucial role in collecting user interactions and quickly understanding the needs of park visitors, highlighting the potential for measuring the impact of implemented solutions. To drive progress, it is essential to prioritize applied research in outdoor environments, placing greater emphasis on implementation rather than ideation.

The process of creating age-friendly environments, as outlined in the WHO cycle, adopts a user-centric strategy akin to Design Thinking [2,90]. However, most of the existing research in this area deviates from this approach. Studies focusing on public parks (e.g., [110,124]), for instance, tend to prioritize solutions without including the crucial step of validation to ensure that the proposed solution effectively addresses the identified problem. To address this gap, the integration of co-design and participatory design methodologies can facilitate a more user-centric approach and enable validation at every stage, from conception to execution and beyond.

The current utilization of physical solutions in public parks typically excludes technological advancements, such as IoT integration in urban furniture and fitness equipment. Incorporating the IoT can enhance the park environment and cater to active aging by gathering valuable data on visitors’ needs and enabling users to access and utilize this information.

Future research initiatives should focus on integrated rather than isolated solutions to advance the field. Existing investigations have predominantly focused on disciplinary solutions, neglecting the interconnected nature of the challenges. A comprehensive strategy encompassing various aspects, such as product design, infrastructure, management, and administration, is required to establish smart public parks that effectively serve older adults. Local councils can contribute to this transition through policy initiatives and incentives, while public funding can serve as a catalyst for innovative approaches. Establishing a structured framework is crucial to form multidisciplinary teams capable of addressing complex challenges and providing guidance to policy-makers and managers. This framework should include an assessment tool to gauge the current level of smartness in public parks and identify essential steps for improvement.

The existing literature emphasizes a significant gap between research-based solutions and market-oriented demands, highlighting the need for a stronger and more interconnected relationship between these two realms [128]. It is crucial to bridge this disparity and foster a synergy between research findings and market-driven solutions to effectively address the needs of various stakeholders. This underscores the significance of fostering synergy and collaboration between research findings and market demands to effectively meet the needs of diverse stakeholders.

The majority of existing qualitative studies (e.g., [107,114,115,120]) have centered on public park users, neglecting the exploration of individuals who have not utilized such facilities. To address this gap, a more comprehensive understanding of the motivations and barriers faced by non-park users is necessary. Furthermore, it is important to recognize that the challenges inhibiting park usage may extend beyond the physical boundaries of the park itself and encompass the surrounding areas.

Developing a comprehensive data collection system is essential to address knowledge gaps and gain in-depth insights into public park usage trends. This will enable more effective analysis and the identification of open solutions and viable paths forward. Regular evaluations are crucial in order to ensure the sustainability of these solutions and accommodate changing user needs and preferences over time, with active involvement in park management.

Once more, incorporating older users throughout the research, design, and implementation phases of smart public parks is of utmost importance. This inclusive approach is necessary to develop design solutions that specifically cater to their unique requirements. By prioritizing the involvement of older adults, we can create an inclusive and accessible public space that ensures their full participation and enhances their experience in the park. This commitment to equity will contribute to the creation of a smart public park that serves the diverse needs of all citizens.

## 7. Conclusions

The main aim of this article was to provide a comprehensive overview of the current views on smart public parks and to underscore the importance of new research in the field to promote active aging. With the older adult population continuing to grow, exploring smart environments that promote active aging has become increasingly crucial. Public parks represent essential urban elements that can positively impact physical and psychological wellbeing. Through the integration of smart technology, public parks can facilitate faster and better decision making for urban planners, while collecting usage data to ensure that implemented solutions are tailored to the population and user needs. By enhancing accessibility and inclusivity, smart public parks can contribute to the development of age-friendly cities while offering opportunities for older adults to participate as active decision-makers.

Aiming for an improvement in the quality of life of older adults through smart public parks is a significant endeavor. By integrating advanced technologies and thoughtful design, these parks could offer a range of benefits tailored to meet the unique needs and preferences of older adults. Therefore, the involvement of older adults in the entire process, from inception to implementation, is crucial. Smart features such as smart lighting, automated seating, and interactive exercise equipment can improve safety, accessibility, and convenience. Multidisciplinary teams should be formed to explore and implement new solutions. It is important to ensure that the technology itself does not burden older adults. For instance, health tracking can be seamlessly integrated into the park environment without requiring the older adult to carry wearables or smartphones. Utilizing non-intrusive sensors and intelligent infrastructure, the smart park system could gather health data discreetly and autonomously, albeit in a manner that respects privacy. This approach ensures that older adults can enjoy the benefits of health tracking without feeling burdened by additional responsibilities or equipment. By considering human factors and adopting a bottom-up approach, technological solutions can be designed to complement the lifestyles of older adults in smart public parks.

The field of smart public parks research presents numerous opportunities, but it remains highly dependent on investments and commitment from decision-makers. Integrated solutions and a framework for low-level implementations are required to facilitate progress. Overall, smart parks could offer a holistic approach to promoting the wellbeing, physical activity, and social engagement of older adults, ultimately enhancing their quality of life in a vibrant and inclusive environment. An approach grounded in systemic design principles may prove instrumental in tackling this multifaceted challenge, which necessitates the collaboration of individuals across various disciplines and sectors while keeping both people and the planet at the forefront of the process [140,141,142]. However, it is important to note that this study did not adhere to a systematic literature review. Thus, it cannot be guaranteed that all relevant information related to smart public parks and age-friendly environments was fully covered.

## Figures and Tables

**Figure 1 geriatrics-08-00094-f001:**
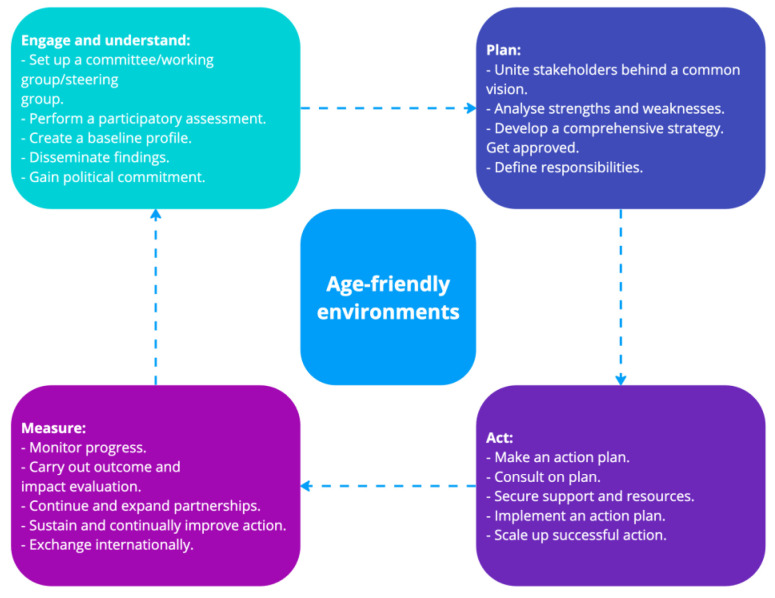
Age-friendly environment cycle of continuous improvement (adapted from [86]).

**Figure 2 geriatrics-08-00094-f002:**
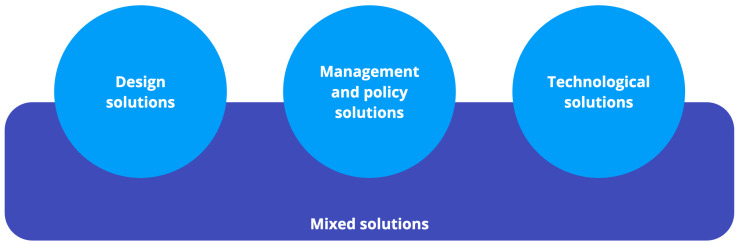
Identified types of solutions.

**Table 1 geriatrics-08-00094-t001:** Age-friendly features related to public parks (adapted from [89]).

Area	Dimension	Age-Friendly Features
Public parks	Buildings and Outdoor Spaces	Age-Friendly Pedestrian System Community Space Availability Inclusive Public Spaces Outdoor Seating Spaces Sign and Way-finding Sufficient and Accessible Public Toilet
	Transportation	Bicycle Strategy Facilities and Amenities at Stop and Station Priority Parking Public Transport to Key Destinations Transportation Option Well-maintained Roads
	Social Participation	Access to Facilities Accessibility of Events and Activities Range of Events and Activities Social Participation and Partnership Social Participation Guide Volunteerism Option
	Respect and Inclusion	Community Inclusion Intergenerational and Family Interaction Recognition and Acknowledgement Respectful and Training
	Civic Participation and Employment	Civic Participation Volunteering Options
	Communication and Information	Access to Communication System Age-Friendly Website Communication Option Information Offer and Delivery
	Community Support and Health Services	Crime-free Emergency Planning Falls Prevention Initiative Health and Support from Social Services Training for Aged People

## Data Availability

No new data were created or analyzed in this study. Data sharing is not applicable to this article.

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
