# Peer review of "Active Aging and Smart Public Parks"

_geriatrics, 2023, doi:10.3390/geriatrics8050094_

Round 1

Reviewer 1 Report

The review addresses a current topic and its major impact on the optimization of the quality of life and the development of proactive behaviors in the elderly by performing physical activities in Smart Public Parks.

Recommendations.

In the Material and Methods section

We recommend highlighting the keywords by which the articles included in this review were selected. We also recommend the addition of a table with the Model for the identification of studies through databases and registers to carry out the extensive review or the Prisma Flow Diagram highlighting the databases used (example: PubMed, Web of Science, Scopus, etc. .

In the Conclusions section

We recommend expanding the conclusions because the most relevant aspects of optimizing the quality of the elderly through in the Smart Public Parks are evident.

No comments.

Author Response

The modifications have been made in Section 5 to emphasize the significance of the database used and the specific keywords employed. However, it is important to note that a systematic literature review or comprehensive review was not conducted, thus the absence of a Prisma Flow Diagram. Moreover, in order to align with the recommendations, the conclusion has been expanded accordingly. 

Reviewer 2 Report

Dear Authors,

At first sight, I was really interested by the abstract and title of your paper.

However, the more I read it, the more I have a difficulty to consider it as a scientific paper. As you will see, I quoted a high number of inappropriate references, which make me consider that this paper is not finished yet, or that this paper is based on falacy because the quotations do not exist.

Furthemore, as the objet of "smart cities" and than "smart parks" is described, one could wait a contradictory (and thus scientific) description of such notions. On the contrary, we only have one side of these notions, i.e. the discursive aspect of them, the "ideal" vision of them. As a social sciences researcher, while this is important, this is totally unsufficient. Even on "parks" it exists a critical litterature ; on "smart cities", the experience of China reveals problematic aspect of indivuduals and groups control. Furthemore, the energy consumption of all technologies, which is a strong critic in a global warming world and time, should also be taken into account, which is not the case.

Here are the complete elements of my argument :

P1 : The migration to large cities has impacted public health. POOR EXPRESSION as migration concerns national movements of population

Obesity, mental illness and lower level of activity :” These issues are inevitable consequences of modern lifestyles as the world becomes more urbanized.

And not in rural areas ? Do you have proof on that ?

P 2 “ Despite efforts to incorporate age-friendly design principles and technologies that pro- 47

mote safety and security, there are still challenges in creating truly inclusive and accessible 48

smart cities and smart public parks for elderly individuals.

THE AUTHOR forget the third pilar of AFCC : participation. And I understand why and this I problematic : smart cities are push driven by technology and industry. Citizens do not ask for more technology.

Furthermore, what is a truly inclusive and accessible city ? Once again, this seems a normative conception of science, not a scientific position.

Indeed, after, it is written : Therefore, it is crucial to address this problem by advocating for the needs of this popula- 52

tion and involving them in the design and development of smart cities and smart public 53

parks, ensuring that their perspectives and needs are fully considered and integrated.

OK But still talking about needs, advocacy and involving them in the design and development arent in line with the citizenship conceptualization of active ageing now developed at WHO, and presented here by Del Barrio et al 2018 : https://www.mdpi.com/2076-0760/7/8/134#:~:text=Active%20Aging%20is%20defined%20as,society'%20(EC%202018).

You seem not to know it.

I also understand that you have a limited vision of the concept of active ageing, relating mainly to physical activity. Indeed, this is correct for some American promotor of this vision in sport and ageing; even WHO support this in the 1990. But since 2002, it is a much more comprehensive vision of ageing underneed.

P 2, L 57-61 : 4 goals are extremely ambitious. Too ambitious in only one paper.  And the section have much too much ambition in their title. A book would not be sufficient to present the 2nd section for ex.

P2 L65 : We shouldnt write anymore the elderly as this is ageist. The common term now is older adults

P2 L69 : In 1946, WHO was the first international body to explicitly link health and well-being The point is not about the date but the fact that WHO is the only international body discussing health issue So this phrase is a non-sense.

P2 L89 : Physical well-being includes the absence of disease, proper nutrition, and exercise.. Is this a quotation or a normative stense ? From empirical qualitative research, there a tons of example that well-being can coexist with disease ; why does it include necesserly exercice ? On the contrary, are all exercices, like for ex. in a constraing job, necessarly good ? No.

Section 3 is the most interesting and well developed part of the paper. However, arent they critical elements about parks and health issues, including exercice? I.e. they might mainly be for higher social classes? Those who are anyway in better health. No ? Mentioning Central park in the paper, here is the argument : https://www.tandfonline.com/doi/abs/10.1080/00222216.1999.11949875

P5, L232 : Is 67 or 64 the source ?

NO DEFINITION of “smart park” in 78, as written.

However the Smart city definition there is very instructive : “A Smart City is an urban area that uses technological or non-technological services or products that: enhance the social and ethical well-being of its citizens; provide quality, performance and interactivity of urban services to reduce costs and resource consumption; and increase contact between citizens and government” (source : https://ieeexplore.ieee.org/stamp/stamp.jsp?tp=&arnumber=8892761, p 8). However, this definition is a problematic according to me as It include the idea of “reducing costs and resource consumption”. Why such a normative vision for a city development ? Why the “reduction of cost” ? It is not a scientific argument, but a political one. Therefore, “Smart cites” are not a scientific concept.

Furthermore, if we consider its idea to “reduce resource consumption”, it is totally contradictory to produce more technology here (https://ecoleurbainedelyon.universite-lyon.fr/resilience-des-vivants-cours-public-par-olivier-hamant-211577.kjsp?RH=ecoleurbainedelyon&ref=dixit.net)

P6, Table 1 : The author inspired by an UN report mentioning the 8 dimensions of AFCC and 64 features. However, they only present 7 and 33 features. Why ? There is no explanation here. Furthemore, I find it very “simple” to duplicate the “public parks” with all the mentioned features. Are these really all valid ? For ex. what does “crime free” or “emergency planning” has to do here ??

P 7 : “Although the age-friendly city and smart city concepts have different goals, they can 278

work toward similar purposes, like greater inclusivity and well-being, through techno- 279

logical advancements ([67], [71] ) ». This quote is a normative one, without a proof. I m not really sure that Tine Buffel and her colleauges (=71) would really say what the author consider they are saying. For me, this example present a unclear and extra-large use of reference, close to the limit of non-scientific norms (assert something, by using a reference that does not say what it is asserted).

P 7 : Modern technology has completely changed how we think about cities. The concept 282

of "Smart Cities" was first introduced in the book "The Technopolis Phenomenon: Smart 283

Cities, Fast Systems, Global Networks," which marked the beginning of the study of 284

"Technological Cities" in the early 1990s [72)  Once again, this quote is wrongly associated with the 72 reference. In this ref., Smart cities are not even quoted ! Only smart growth”… Nothing to do. So, once again, this is wrong science or Falacy.

P7 : And again… “According to [73],smart cities are places where technological innovations are used to enhance urban management and productivity. [73] defines a "smart city" as a location where traditional net works and services are improved to benefit its citizens and businesses using digital and 292

telecommunication technologies. » AGAIN, a FALSE QUOTATION. 73 never wrote this !! (Furthermore, the 73 ref. in not complete in bibliography)

Section on “smart cities”. The authors only present the “discursive” vision of this notion. Critical work exist on the problem of “smart cities” as it is now clearly demonstrated in China where the freedom of movement of the individuals is totally under control by the technology. Another critique regards the energetic cost of such cities. In a time of global warming, this brings more questions than answers.

P9 : “[78] defined a smart park as "a park that uses technology to achieve a series 353

of values: equitable access, community fit, enhanced health and safety, resilience, water, 354

and energy efficiency, and effective operations and maintenance." » FALSE : The 78 author do not mention « park » in their paper.

Based on all these false reference, I can not longer evaluate this paper as I consider it a work made of fallacy regarding the sources. Due to the non critical / reflexive presentation of “smart cities” (an thus “smart parks”) I m simple not able to continue to estimate this work as a proper scientific paper.

Yours sincerelly,

Author Response

First and foremost, we would like to draw attention to an issue regarding the numbering of references in the manuscript. Regrettably, at a certain point, all the references were incorrectly numbered. We have already addressed this matter with the editor and apologize for any confusion caused. In the latest version of the manuscript, all references have been rectified and are accurately numbered.  

P1. More detail and clarification are added to the information about city mitigation.  

 P2. Because we believe that participation is essential to creating new types of cities, we emphasize the third pillar of smart cities—"participation"—throughout the entire manuscript. The active aging section is expanded to provide more information on the topic. Each introduction section's goal is examined and revised. "Older adults" is used instead of the word "elderly". More references are added to Section 4.2 (Age-friendly and Smart City) and Section 2.2 (Health and Well-being).   

P5. More references are included to illustrate various points of view because there is no consensus on the definition of smart cities.   

P6. The rationale for presenting 7 rather than 8 dimensions is clarified.   

P7. The final paragraph of Section 4.1 on age-friendly cities has been revised and expanded.   

P9. More references are cited in Section 4.3 (Smart Public Park), which has been improved.   

Reviewer 3 Report

Here are my comments:

There is no definition offered in either abstract or introduction to define smart cities or smart parks, critical to understanding the passages and article overall. Definition begins on line 282.

Introduction would benefit from data on chronic disease rates overall.  The concept of smart parks to reduce or delay chronic disease prevalence should be added. Excellent discussion of urban health issues noted.

Paragraph starting on line 175 is disjointed.  The first three lines don’t tie to line four’s “therefore” statement.  Please amend.

Please change line 191 and 192 - Avoid “no cure for aging” and correct “steps to speed up the aging process” show be “slow aging process”.

Section on Age-friendly cities beginning on line 220 should mention Age-Friendly Health Systems, rounding out the WHO references with the importance of design of healthcare delivery within Age-friendly cities (consider adding reference from Terry Fulmer.

The section beginning on line 281 is innovative and a significant contribution to the field.

Line 440 great point about benches promoting social networks but consider adding their role in promoting safety.

Line 523 sounds ageist - reword.

Line 542 to 544 is a false dichotomy - should be an “and” statement ie live long AND live well.

Only comment offered - “older adults” preferred over use of word “elderly”.

Author Response

Every aspect of the recommendations has been addressed, with the exception of the definition of smart cities or smart parks in the introduction/abstract. Due to the absence of a consensus on the exact definition of these concepts, multiple references are used throughout the manuscript to define both smart cities and smart parks.  

Round 2

Reviewer 2 Report

Dear Authors,

Thank you for all your modifications.

I just suggest some last changes ; while they are not many, they are still major.

I suggest to delete all "truly" notions, as it is too normative, with no empirical proof.

p5 : I should delete this phrase : "While there is no cure for aging, promoting active aging can play a significant role in maintaining overall wellbeing." because "aging" is not an illness to be cured.

p5 : It is historically wrong to say that Kalache introduced the concept of AA. It is correct to say he played an important role in promoting it in a comprehension version ; but Alan Walker also played a certain role in promoting AND theoritically building it (while Kalache did not make theory of it.) There are a series of critical authors who informed on that historical origins : Avramov/Mascova ;  Boudiny Kim ; Biggs Simon/Moulaert thibauld;  Lassen Aske Juul ; van Dyk Silke ; Timonen Virpi .

Not only WHO promotes Active aging ; OCDE did it in 1998 and EC (european commission) did it since 1999 too.

Yours sincerelly

OK

Author Response

Last document version without the word "truly".